# Statistical guidelines for quality control of next-generation sequencing techniques

Maximilian Sprang ⬤, Matteo Krüger, Miguel A Andrade-Navarro ⬤, Jean-Fred Fontaine ⬤

**More and more next-generation sequencing (NGS) data are made available every day. However, the quality of this data is not always guaranteed. Available quality control tools require profound knowledge to correctly interpret the multiplicity of quality features. Moreover, it is usually difficult to know if quality features are relevant in all experimental conditions. Therefore, the NGS community would highly benefit from condition-specific data-driven guidelines derived from many publicly available experiments, which reflect routinely generated NGS data. In this work, we have characterized well-known quality guidelines and related features in big datasets and concluded that they are too limited for assessing the quality of a given NGS file accurately. Therefore, we present new data-driven guidelines derived from the statistical analysis of many public datasets using quality features calculated by common bioinformatics tools. Thanks to this approach, we confirm the high relevance of genome mapping statistics to assess the quality of the data, and we demonstrate the limited scope of some quality features that are not relevant in all conditions. Our guidelines are available at https://cbdm.uni-mainz.de/ngs-guidelines.**

## Introduction

Next-generation sequencing (NGS)–based analyses of regulatory functions of the genome are widely used in clinical and biological applications and have gained a key role in research in recent years. Many different assays have been developed reaching from the classical sequencing to gene expression quantification (RNA-seq), identifying epigenetic modifications (ChIP-Seq, Bisulfite Sequencing) and measuring chromatin accessibility (DNAse-seq, MNAse-seq, and ATAC-seq).

NGS experiments require stepwise data analysis, to gain information from short reads, which first need to be assembled or aligned to a reference genome. It is crucial to filter out low-quality data as early as possible to prevent negative impact on downstream analysis (1, 2). Especially in the clinical context, misinterpretation of data due to faulty samples can have dire consequences

for patients, such as false diagnosis or wrong therapy approaches. We could show in a previous work that the systematic removal of lower quality samples within RNA-seq datasets improves the clustering of disease and control samples (3).

There are a variety of tools that can be used to compute features holding information about the quality of NGS data. Classical quality control (QC) tools analyze raw data exported from the machine performing the assay. The raw data are stored in FastQ files, which contain the sequence of the read and a corresponding quality score Q, encoded in ASCII characters. The score Q is an integer mapping of the probability $P$ that a base call is incorrect (4). Manual interpretation of these scores is not possible because each base of each read must be taken into account.

The most popular tool for quality control of FastQ files is FastQC (5), which can be used to get multiple features that hold information on the quality of the raw data. Examples are position-dependent biases, sequencing adapter contamination and DNA over-amplification. Downstream analysis can also provide insight on the quality of the given data. For example, genome mapping statistics include the number of reads mapped to a unique position or unmapped reads, which are significant with respect to the quality of the input data (6, 7, 8, 9). While raw quality score and mapping statistics can be used in combination with any NGS-sequencing data, for some applications additional steps can be taken to complement the quality analysis of the data. For chromatin and protein-DNA interaction assays, such as DNAse-seq, ATAC-seq and Chip-seq, it may be of interest for quality evaluation to use the genomic locations and the distribution of reads near transcription start sites (TSSs), which are of interest in these assays anyway (10, 11, 12, 13).

Applying all these methods and using their features to determine the quality of a new sample can be complicated. The information about quality held by some features could substantially vary depending on the assay as shown by the popular experiment guidelines from the ENCODE project (14). For example, these guidelines recommend a minimal number of uniquely mapped reads to evaluate DNA-seq sequencing files, while the number of useable fragments should be considered for ChIP-seq with different thresholds for narrow-peak or broad-peak experiments. Given in addition to specific good wet-laboratory practices, those threshold values alone may not be able to accurately classify

Faculty of Biology, Johannes Gutenberg-Universität Mainz, Biozentrum I, Mainz, Germany

Correspondence: fontaine@uni-mainz.de

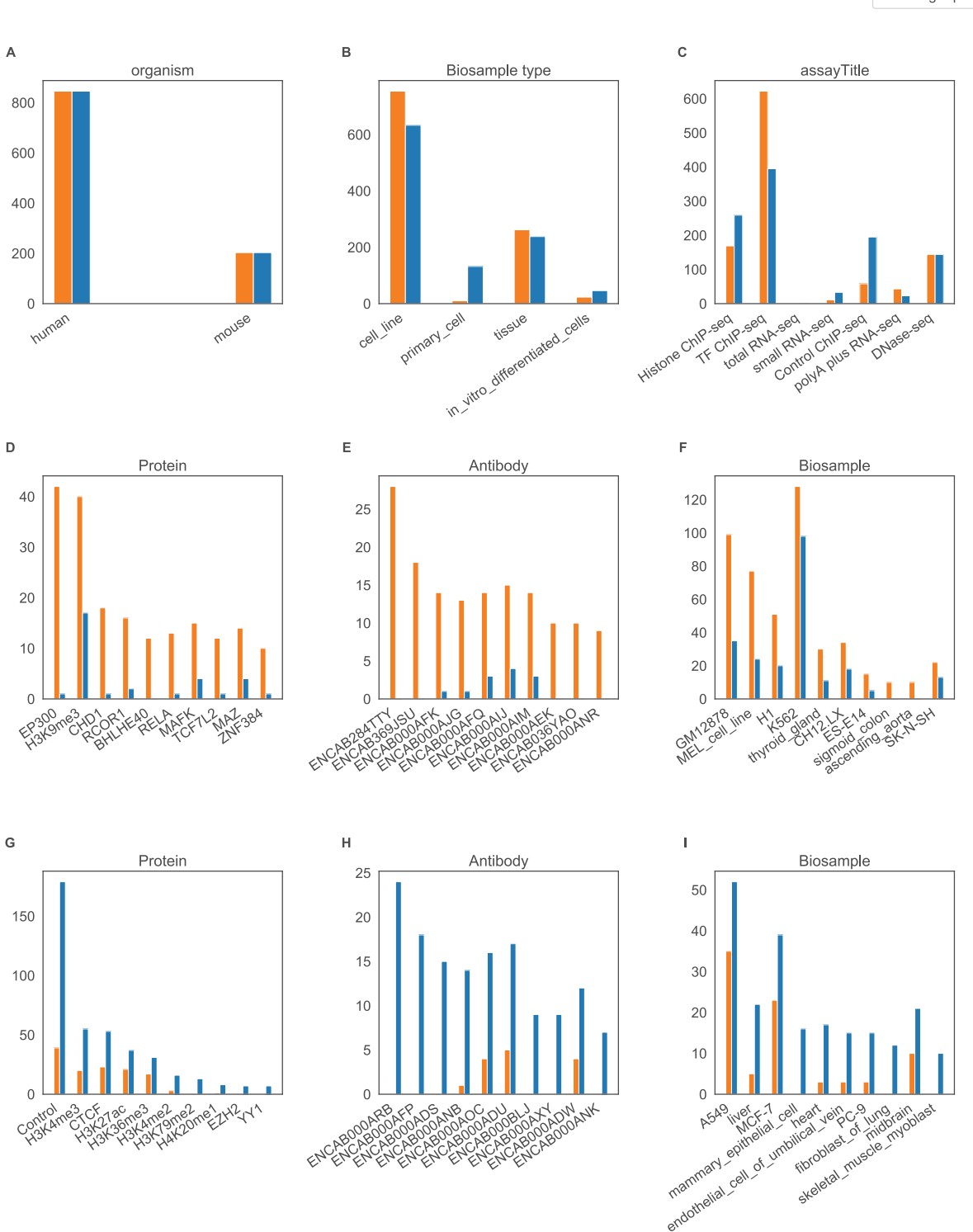

**Figure 1. Distribution of low- and high-quality files for different experimental parameters in the dataset.**
**(A, B, C)** Distribution of files by organism, biological sample type, and assay in the dataset. **(D, E, F)** Distribution of files by ChIP protein, ChIP antibody and biological sample (only the 10 files with the biggest difference between high- and low-quality are shown). **(G, H, I)** Distribution of files by ChIP protein, ChIP antibody and biological sample (only 10 most frequent annotations in low-quality files shown). There was a total of 269 proteins, 349 antibodies and 212 biological sample types.

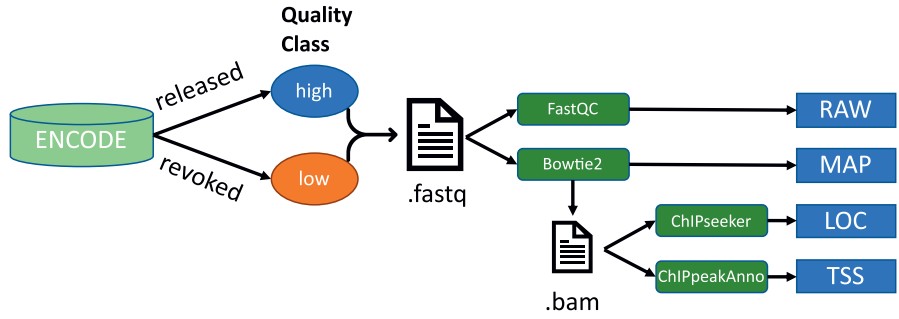

**Figure 2. Methods to identify features of possible importance towards assessing the quality of a file.**
High- and low-quality files are defined using their label in ENCODE (released and revoked, respectively). The raw FastQ files are used as input for FastQC and Bowtie2 tools, respectively returning RAW and MAP feature sets. The mapped reads from Bowtie2 are used as input for ChIPseeker and ChIPpeakAnno packages, returning the LOC and MAP feature sets, respectively.

sequencing files by quality, and unfortunately there are no guidelines for specific experimental conditions, such as RNA-seq in liver cells or CTCF ChIP-seq in blood cells. Other large-scale repositories of NGS files may propose their own guidelines, such as the Cistrome project, which defines thresholds for six quality features to differentiate low- and high-quality files. Yet, in the Cistrome data portal (15), the relevance of the features under specific conditions and the best combination of features to make a final decision remain unclear.

Taken together, using all available tools and guidelines, it remains challenging or impossible to know the relevance of each quality feature under different experimental conditions and to know what combination of features and threshold values would provide an accurate classification of files by quality.

To better define the relevance and scope of application of quality features, we have used the ENCODE repository, which contains a large number of high- and low-quality NGS data that have been labelled as released or revoked, respectively (16, 17). The ENCODE curators manually decided the quality of submitted files after reviewing multiple quality features (18, 19). We used 2,098 curated files and focused our work on four types of features: a first set from the analysis of raw quality scores, a second set from genome mapping statistics, and two more sets from downstream analysis tools that call peaks in the mapped data (genomic localization and TSS-relative position). As the files have been generated using common sequencing protocols in various laboratories, they reflect routinely produced data. More specific protocols are not included in this work although they may be relevant to study quality (e.g., recent PCR-free methods). After the evaluation of the current ENCODE and Cistrome guidelines, we studied the relevance of the selected quality features by deriving their discriminative and classifying power. Finally, we derived machine learning–based decision trees and created a public web interface to explore our data-driven guidelines.

## Results

### Assessment of available quality information

#### ENCODE
As part of its general NGS guidelines, the ENCODE project has published some numerical guidelines for the release of NGS experiments in their repository. They are defined as a minimal number of uniquely mapped reads, aligned reads, or useable fragments in different assays. To assess the relevance of these guidelines, we compared them with actual quality features derived from many ENCODE data files in relation to the manually annotated quality status of those files (see Fig 1 for an overview of the data, Fig 2 for the pipeline that derives quality features and Fig 3 for the comparison).

The distributions of aligned reads are depicted for H3K9me3 ChIP-Seq and RNA-Seq files in Fig 3A. For H3K9me3, the low-quality files do have in general smaller numbers of aligned reads, but half of the high-quality files do not reach the given guideline of at least 45 million reads (blue dashed line). For RNA-seq, the guideline at 30 million reads (red dashed line) cannot differentiate between high- and low-quality files, as more than 75% of each type of files is above the value.

For DNase-Seq, ENCODE provides a minimum of 20 million uniquely mapped reads as a guideline for good quality (orange dashed line in Fig 3B). Low- and high-quality files have similar distributions of uniquely mapped reads and the guideline matches the median value. Hence, the number of uniquely mapped reads does not seem to offer reliable information about the quality of the files.

The distributions of the number of useable fragments for good and bad quality ChIP-seq files display more differences for files related to broad peak experiments than for files related to narrow peak experiments (Fig 3C). Whatever the file quality, ENCODE guidelines are either seldom met (broad peak data) or always met (narrow peak data).

Overall, the numerical ENCODE guidelines alone cannot be seen as a reliable measurement for quality. Yet, the features they are based on may be more useful if integrated in specific classification algorithms or if applied to data subsets defining particular experimental conditions.

### Assessment of available quality information

#### Cistrome
Another source of NGS data with quality information is the Cistrome database, which gathers NGS data for DNA–protein interaction and chromatin accessibility assays (15). Cistrome computes six quality features using their own quality control pipeline (20) and defines six corresponding threshold values, considered here as guidelines, to flag good or bad quality files if the feature is greater or lower than the threshold, respectively.

The distributions of the feature values depending on the number of low-quality flags found for the other features are shown in Fig 4A for the complete set of available data files in Cistrome. We can observe that guideline values, in general, poorly separate low- and

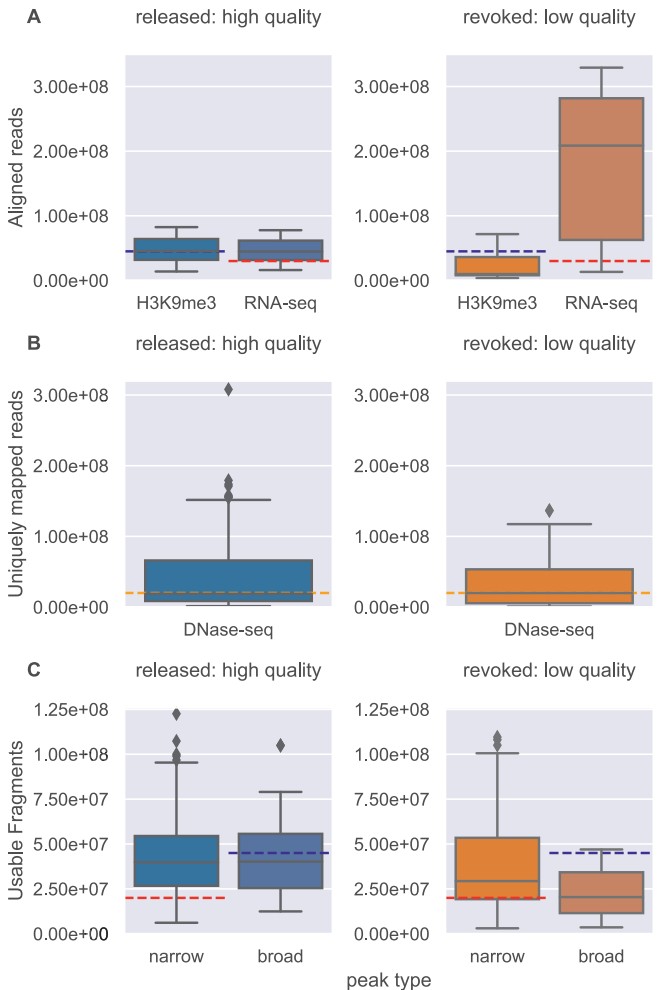

**Figure 3.   Comparison of the manually curated quality and the guidelines given by ENCODE for high-quality files.**
Horizontal dashed lines are minimal values given as guidelines. Guidelines as well as files are all from ENCODE version 3 (2013–2018). **(A)** Number of aligned reads for H3K9me3, ChIP-Seq, and RNA-Seq. The blue dashed line denotes 45 million reads given as minimal guideline for H3K9me3 ChIP-Seq and the red line denotes 30 million reads given as minimal guideline for RNA-Seq. **(B)** Uniquely mapped reads for DNAse-Seq data. The orange dashed line indicates the ENCODE guideline of 20 million reads. **(C)** Useable fragments for narrow and broad peak ChIP-Seq. The blue dashed line is at 45 million and the red dashed line at 20 million, indicating ENCODEs guidelines for broad and narrow peak data, respectively.

high-quality files. Yet, the uniquely mapped ratio seems to have some discriminative power for files associated with four bad flags.

In addition to comparing distributions, we computed pairwise Pearson's correlation coefficients between the features (Fig 4B) and Pearson's, Spearman's and Kendall's correlation coefficients between the features and the number of additional bad flags (Fig 4C). Pairwise correlation (Fig 4B) indicates that the three more complex features, which correspond to peak analysis, correlate positively with each other. The strongest correlation (r = 0.54) between fraction of reads in peak (FRiP) and PeaksFoldChangeAbove10 could be expected because a high FRiPs is likely to lead to more peaks with high fold change. Regarding the correlation of features with number of low-quality flags of the other features (Fig 4C), for an

informative feature, we would expect a negative coefficient. Although coefficients were negative, they were moderate in absolute value (<0.33). Stronger correlation coefficients were associated with the more complex features. PCR bottleneck coefficient (PBC) showed virtually no correlation.

Taken together, Cistrome's guidelines do not offer a powerful solution to differentiate low- and high-quality data files: related quality features are mostly inconsistent with each other (no strong correlation with the number of bad flags from other features) and features with highest potential are partly redundant with each other.

### Relevance of quality features in experimental conditions

According to the ENCODE metadata of a total of 2,098 files, high-level subsets that grouped files by organism, assay, and run type could be highly biased towards a few biological samples or, for ChIP-seq, a few protein or antibody targets (e.g., see subset mouse single-end control Chip-seq on page 16 of Supplementary_trees.pdf file [Supplemental Data 1]). Therefore, to define more practical guidelines that would take into account possible differences between experimental conditions, we focused our analysis on lower level subsets and derived 47 quality features (see Fig 1 and Table S1) from the related NGS files to study their ability to differentiate low- and high-quality files.

We defined subsets of data files in three groups by unique combinations of the following parameters from the metadata:

- Group A: subsets defined by assay title, organism, run type and biological sample.
- Group B: subsets defined by assay title, organism, run type and protein target.
- Group C: subsets defined by assay title, organism, run type, protein target and antibody.

The number of all subsets is 436 for subset group A, 354 for B and 461 for C. We focused our analysis on subsets with at least 10 files: 38 subsets in group A, 41 in B and 33 in C. We could not study a group gathering all parameters of groups A and B or C because it resulted in only one subset with at least 10 files.

Within each of the three groups of subsets and considering a given feature, differences could be observed between subsets. For example, in group A, for subsets related to mouse paired-ended (PE) DNAse-seq, the percentage of reads located between +4,000 and +5,000 bp relative to TSSs (TSS_+4500) was significant for brain tissues but not for limbs (Table S2); in group B, for subsets related to mouse single-ended (SE) histone ChIP-seq, the percentage of reads that could be mapped to the reference genome (MAP_MI_over-all_mapping) was significant for all histone marks but not for H3K36me3 (Table S3); in group C for subsets related to CTCF mouse SE ChIP-seq, the percentage of reads that were mapped to multiple locations of the reference genome (MAP_MI_multiple_mapping) was significant for antibody ENCAB210NHK but not for antibody ENCAB000AFQ (Table S4).

To enable observation of quality features for any experimental condition in our dataset, we have implemented an interactive dashboard publicly available at https://cbdm.uni-mainz.de/ngs-guidelines. From this web interface, users can select specific experimental

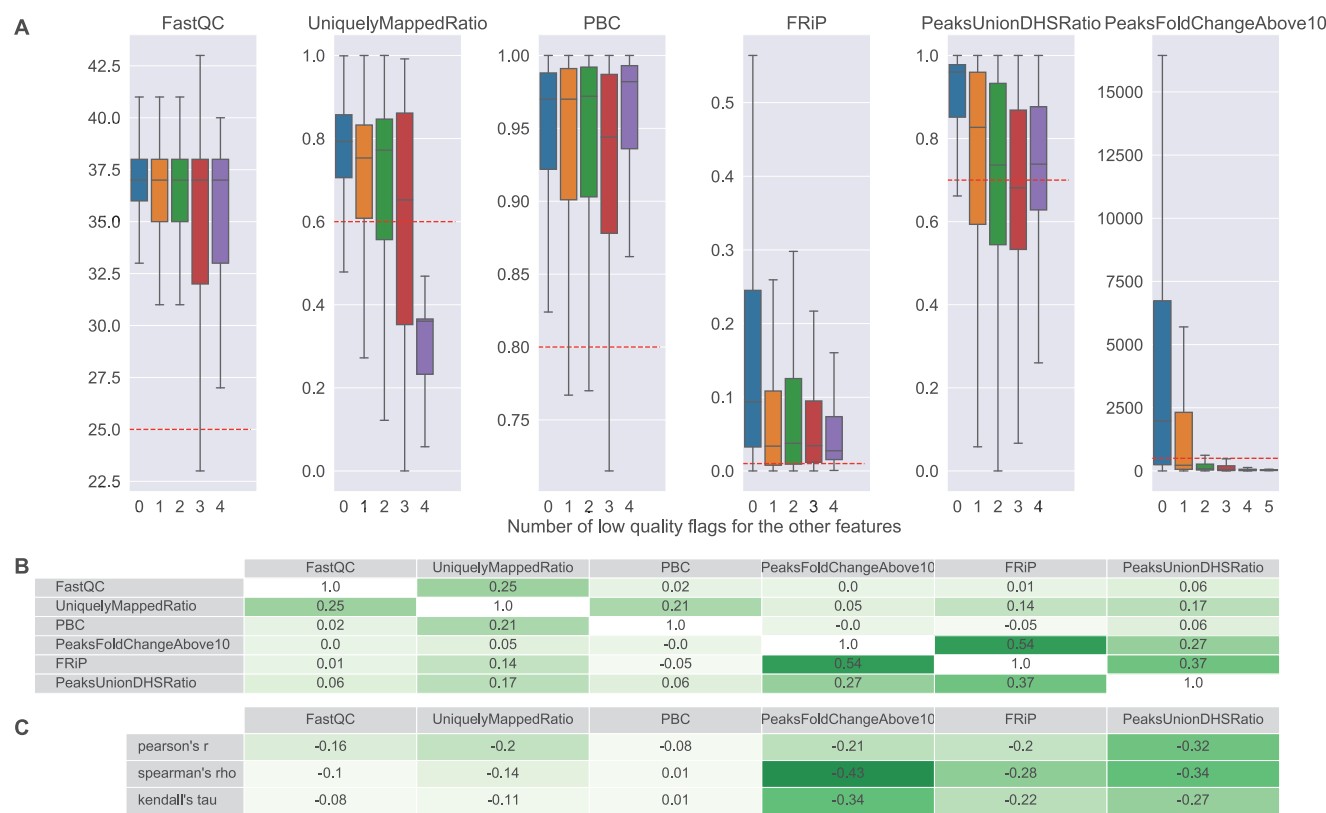

**Figure 4. Six features used for quality assessment in the Cistrome database.**
**(A)** From left to right: FastQC's raw sequence median quality score (5), uniquely mapped reads ratio of BWA's mapping (7), PCR bottleneck coefficient (PBC), fraction of reads in peaks (FRiP) (17), proportion of the 500 most significant peaks overlapping with a union of DNase-seq peaks derived from ENCODE files (PeaksUnionDHSRatio) (20, 34), and number of peaks called by MACS2 with a fold change above 10 (PeaksFoldChangeAbove10). The thresholds for the features are: 0.25, 0.6, 0.8, 0.01, 0.7, and 500, respectively. The x-axis shows how many quality flags were indicating low quality excluding the flag of feature being represented. Boxplots do not show outlier points. **(B)** Pairwise Pearson's correlation coefficients of Cistrome's features. **(C)** Pearson's, Spearman's and Kendall's correlation coefficients of each Cistrome's feature with the number of low-quality flags of the other features.

conditions to reproduce our subsets or to visualize statistics for custom higher- and also lower-level subsets (Fig 5). The three tabs in the web interface, from left to right, are dedicated to detailed statistics on each feature, assay and antibody, respectively. The first tab was used to create Fig 5 and is especially suited to assess a file's quality using selected features. The assay tab gives an overview of the quality features and the underlying data of user-defined subsets. The antibody tab is not only useable for quality classification, but can also be used to search for antibodies that have been used for a target of interest; if there is information about multiple antibodies a user can pick the one with the best performance (see Fig S1).

## Power of each quality feature

To assess the power of each quality feature in each data subset, we first performed statistical tests to know if average feature values per quality are significantly different. We also calculated areas under Receiver Operating Characteristic curves (auROC) to assess classification power of the features in the data subsets.

Results of the statistical tests are summarized in Fig 6, which shows the number of subsets in which each feature was found to be

significant to differentiate files by quality for the three groups of subsets and three different false discovery rate (FDR) thresholds. The genome mapping features set (MAP) dominate the plot for all FDR thresholds. Raw reads statistical features (RAW) are also frequently found, especially for the lowest FDR threshold. Overall, 21 of the 47 features appear in the top 10 selected for each group and the three FDR thresholds. FDR values computed for each subset and each quality feature can be seen in the interactive online dashboard. At the bottom of the view, there is a table containing subsets ordered by descriptive parameters and respective FDRs.

Table 1 shows the average auROC value for each feature, calculated across each subset containing at least 10 files in the same group. The confidence of the classification based on one feature differed greatly depending on the subsets.

Most notably, the mapping features have all high predictive power with auROC values up to 0.86. Interestingly, many of FastQCs features have low predictive power or are even close to random (auROC = 0.50), such as sequence length distribution statistics. The difference in classification power of paired-end features for the mapping statistics (MAP_PE) between subsets from group A and the others, is likely due to the fact that more than half paired-end files are DNAse-Seq files, which are only found in group A.

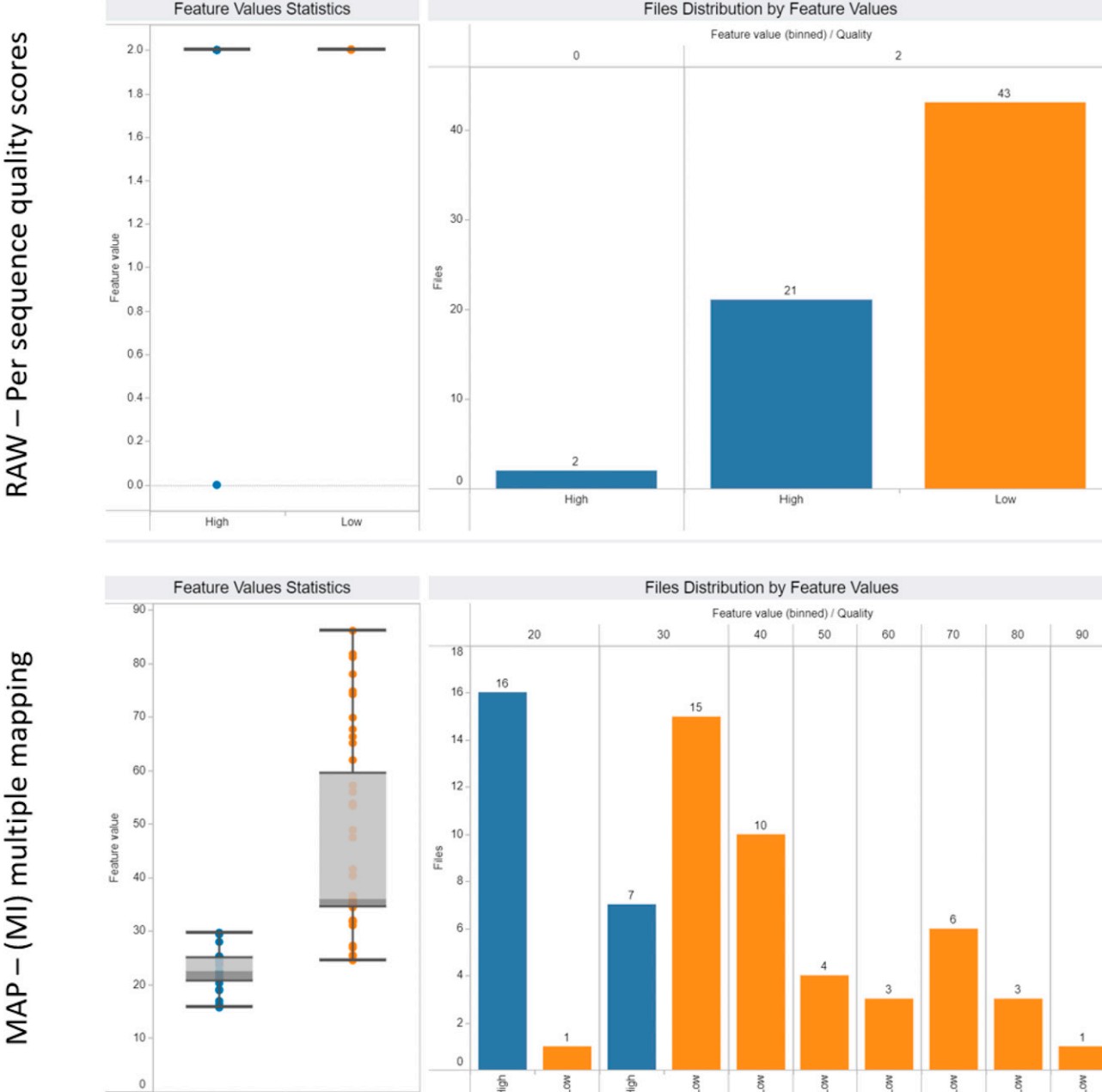

**Figure 5.  Comparing features in custom subsets using the dashboard.**
The data were filtered for polyA plus RNA-seq files, a higher level data subset compared to subsets defined for groups A, B, and C. Two example features were selected: per sequence quality score (top row) and multiple mapping (bottom row). On the left-hand side, boxplots show the distributions of values for high- (blue) and low-quality (orange) files for the respective features. On the right-hand side, the histograms of values are shown. Yet, using the dashboard, we can conclude that the multiple mapping feature is more powerful than per sequence quality score to differentiate files by quality.

Contrary to statistical tests, the auROC calculations do not take into account the sample size and variance. Therefore, the reliability of individual values is limited for the many small-sized subsets in this analysis but median values as shown in this summary may be seen as a more robust evaluation. This robustness is suggested by the agreement with the statistical tests, which also find MAP features the most powerful. Results for higher level subsets are available as Supplementary information (see Fig S2 and Table S5 for group 1 and Fig S3 and Table S6 for group 2).

## Decision trees as practical guidelines

We derived practical guidelines for NGS scientists by using decision trees combining multiple features for classifying files by quality in our data subsets (Supplementary_trees.pdf [Supplemental Data 1]). These trees can be used for a more illustrated view on how to use our features as guidelines. Fig 7 shows two examples built during the learning phase of a CART decision tree (21). A first example tree

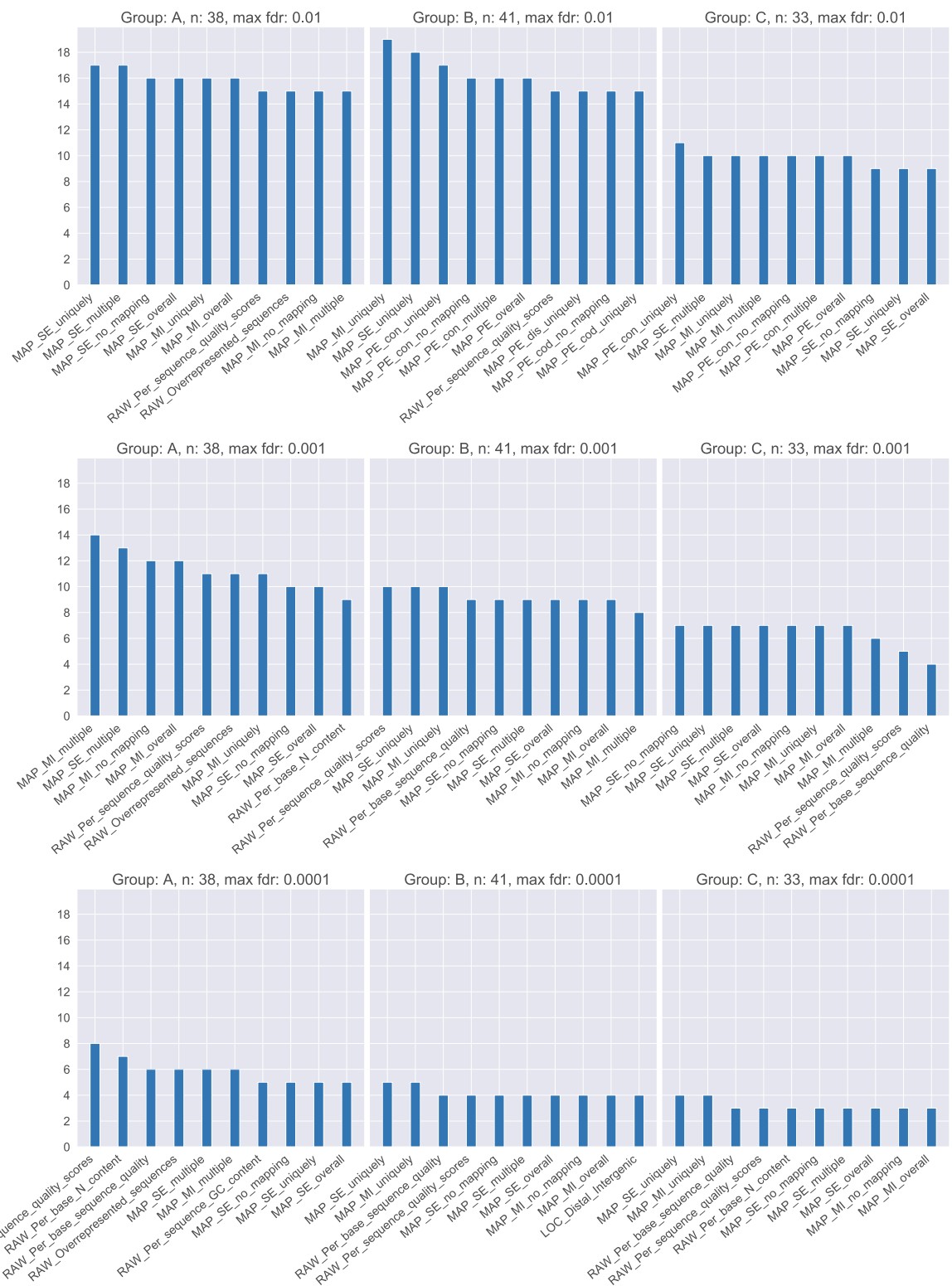

**Figure 6. Top features most often significant to differentiate files by quality in each of the three subset groups.**
For each of the three subset groups (A, B, and C), the number of subsets (y-axis) in which a quality feature is significant is shown. Subsets have a minimum of 10 files and the *P*-value cutoff for significance is defined as either 0.01 (top row), 0.001 (middle row) or 0.0001 (bottom row). The total number of subsets in each subset group is 436 for subset group A, 354 for B and 461 for C. The number of subsets with at least 10 files in each subset group is 38 for subset group A, 41 for B and 33 for C.

**Table 1.  Average classification performance of the quality features in each subset group.**

| Quality feature | Group A | Group B | Group C |
|---|---|---|---|
| RAW_Basic_Statistics | 0.500 | 0.500 | 0.500 |
| RAW_Per_base_sequence_quality | 0.681 | 0.738 | 0.747 |
| RAW_Per_tile_sequence_quality | 0.645 | 0.668 | 0.658 |
| RAW_Per_sequence_quality_scores | 0.693 | 0.738 | 0.754 |
| RAW_Per_base_sequence_content | 0.650 | 0.622 | 0.632 |
| RAW_Per_sequence_GC_content | 0.684 | 0.686 | 0.652 |
| RAW_Per_base_N_content | 0.674 | 0.732 | 0.745 |
| RAW_Sequence_Length_Distribution | 0.502 | 0.523 | 0.511 |
| RAW_Sequence_Duplication_Levels | 0.606 | 0.586 | 0.602 |
| RAW_Overrepresented_sequences | 0.771 | 0.763 | 0.740 |
| RAW_Adapter_Content | 0.551 | 0.526 | 0.538 |
| RAW_Kmer_Content | 0.559 | 0.574 | 0.549 |
| MAP_SE_no_mapping | 0.840 | 0.817 | 0.841 |
| MAP_SE_uniquely | 0.829 | 0.821 | 0.850 |
| MAP_SE_multiple | 0.860 | 0.805 | 0.859 |
| MAP_SE_overall | 0.840 | 0.817 | 0.841 |
| MAP_MI_no_mapping | 0.837 | 0.809 | 0.825 |
| MAP_MI_uniquely | 0.839 | 0.817 | 0.841 |
| MAP_MI_multiple | 0.847 | 0.798 | 0.841 |
| MAP_MI_overall | 0.846 | 0.810 | 0.827 |
| LOC_Promoter | 0.719 | 0.717 | 0.717 |
| LOC_5_UTR | 0.706 | 0.692 | 0.687 |
| LOC_3_UTR | 0.745 | 0.711 | 0.725 |
| LOC_1st_Exon | 0.699 | 0.688 | 0.702 |
| LOC_Other_Exon | 0.733 | 0.724 | 0.733 |
| LOC_1st_Intron | 0.687 | 0.716 | 0.705 |
| LOC_Other_Intron | 0.686 | 0.717 | 0.697 |
| LOC_Downstream | 0.681 | 0.688 | 0.697 |
| LOC_Distal_Intergenic | 0.727 | 0.719 | 0.710 |
| TSS_−4500 | 0.690 | 0.692 | 0.676 |
| TSS_−3500 | 0.696 | 0.707 | 0.710 |
| TSS_−2500 | 0.682 | 0.694 | 0.699 |
| TSS_−1500 | 0.702 | 0.685 | 0.708 |
| TSS_−500 | 0.703 | 0.731 | 0.723 |
| TSS_+500 | 0.706 | 0.718 | 0.718 |
| TSS_+1500 | 0.686 | 0.702 | 0.724 |
| TSS_+2500 | 0.691 | 0.709 | 0.722 |
| TSS_+3500 | 0.677 | 0.703 | 0.712 |
| TSS_+4500 | 0.691 | 0.692 | 0.701 |
| MAP_PE_con_no_mapping | 0.833 | 0.711 | 0.711 |
| MAP_PE_con_uniquely | 0.852 | 0.772 | 0.772 |
| MAP_PE_con_multiple | 0.830 | 0.708 | 0.708 |

*(Continued on following page)*

**Table 1.  Continued**

| Quality feature | Group A | Group B | Group C |
|---|---|---|---|
| MAP_PE_dis_uniquely | 0.771 | 0.676 | 0.676 |
| MAP_PE_cod_no_mapping | 0.837 | 0.626 | 0.626 |
| MAP_PE_cod_uniquely | 0.773 | 0.673 | 0.673 |
| MAP_PE_cod_multiple | 0.849 | 0.655 | 0.655 |
| MAP_PE_overall | 0.853 | 0.724 | 0.724 |

Classification performance is measured as area under Receiver Operating Characteristic curve (auROC) from 0.5 for a random classification and 1.0 for a perfect classification. MAP features perform best over all three groups. The RAW features that perform well do so over all three groups. For LOC and TSS only some of the features show good performance on average; however, these still can be more important for some of the subsets in each group.

related to human single-end H3K4me1 ChIP-seq needs the maximum depth of three splits to perfectly classify the files (accuracy = 100%). The tree first splits the data at a value of 15.68% for the percentage of multiply mapped reads (MAP_SE_multiple) by checking which files have values that are smaller or equal to the threshold (top node). This holds true for nine files, which are mostly of low quality (left branch). The next split on the left branch uses the percentage of reads located in non-first introns (LOC_Other_Intron) at a value of 34.072%; this time, all seven files below the threshold are of low-quality (first leaf from the left) and the other files of high-quality (second leaf). From 21 files on the right branch, 13 high-quality files (last leaf) have a percentage equal or <2.466% of reads located between +2,000 and +3,000 bp of TSS regions (TSS_+2500). The remaining eight files are further divided in either five low-quality files that are associated with a percentage equal or <3.701% of reads located between +0 and +1,000 bp of TSS regions (TSS_+500), or three high-quality files otherwise. A second example tree related to human single-end CTCF ChIP-seq on Fig 7 needs only two splits to also perfectly classify the files (accuracy = 100%). The performance of all the trees on their training set is summarized in Table S7. Results for higher level subsets as well as trees for all subsets, including many trees that require only one split node to achieve 100% accuracy, are given in the supplementary material: Supplementary_trees.pdf (Supplemental Data 1).

## Discussion

Although next-generation sequencing has been in use for years, certainty in quality control can still not be achieved (22). We assessed existing quality guidelines (15, 18) and implemented data-driven approach with a set of features derived from different common QC and analysis tools because existing guidelines could not be used to safely distinguish high- from low-quality NGS files. The popular ENCODE guidelines offer a standard to guide the production of new results, but do not accurately represent the curated labels provided by the ENCODE's internal quality control procedure (Fig 3), which were used as quality classes for this work. Cistrome is another valuable repository of functional genomics data that also provides some features that give information about quality. However, they

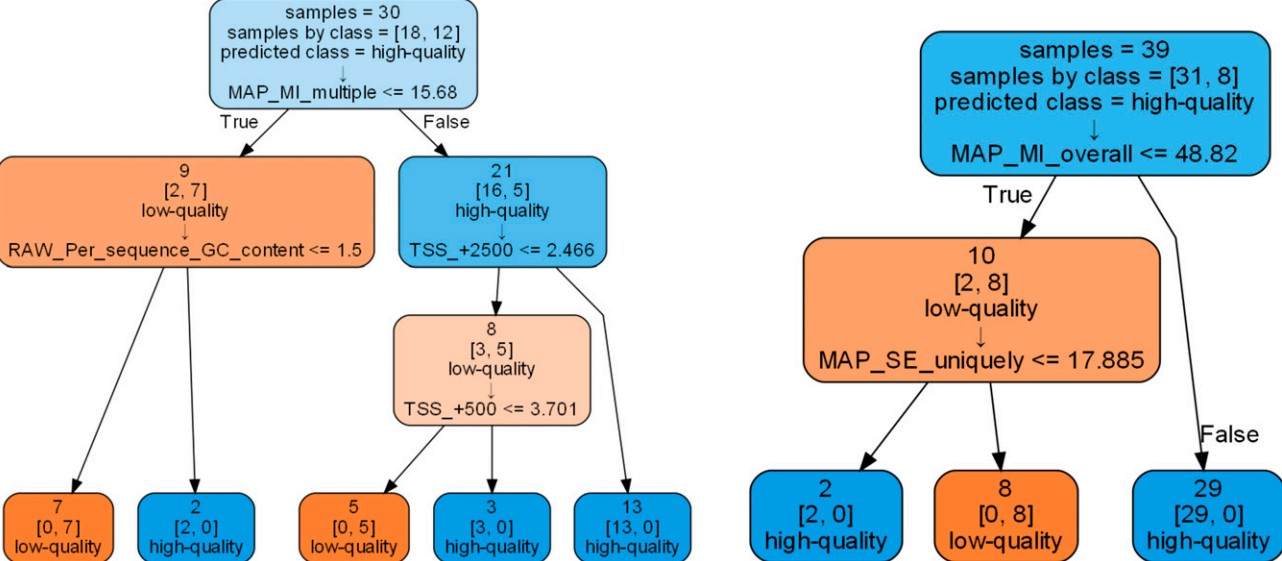

**Figure 7. Decision trees derived with the CART algorithm.**
The Gini-criterion is used and a maximum depth of three was set. The two decision trees are related to human single-ended (SE) H3K4me1 ChIP-seq (left-hand side; p58 in Supplemental Data 1) and human SE CTCF ChIP-seq (right-hand side, p64 in Supplementary trees file [Supplemental Data 1]) and achieve 100% accuracy to classify related files by quality. Every split node contains the number of files in the node (samples) and the files' true classification of quality (samples by class = [high-quality files, low-quality files]), as well as the prediction for this node (predicted class). At the bottom of every node, the quality feature that will be used for the split and the corresponding threshold are given. MAP_SE_multiple: percentage of reads that are mapped to multiple genomic locations in a SE experiment; LOC_Other_Intron: percentage of reads in non-first intron regions; TSS_+2500 and TSS_+500: percentage of reads in [2,000, 3,000] or [0, 1,000] bp region, respectively, relative to transcription start sites; MAP_SE_no_mapping: percentage of reads that could not be mapped to reference genome in a SE experiment; LOC_Distal_Intergenic: Percentage of reads in distal intergenic regions.

proved not to be practical enough (Fig 4). Moreover, guidelines derived from either ENCODE or Cistrome do not take into account potential differences between cell types or ChIP protein and antibody targets that could have an impact on quality metrics. Therefore, we derived data-driven guidelines by studying 47 selected features in high- and low-level subsets of 2,098 manually labelled data files, covering experiments specific to particular biological samples, ChIP protein targets or antibodies. We observed that in higher levels of the data, when dividing the data only by organism, runtype and assay, good classification results should be carefully checked against possible biases in data composition (e.g., over-representation of a few cell types). Therefore, we divided our dataset further into smaller less-biased subsets and found that some features differ significantly in their discriminative power when different cell types or ChIP protein and antibody targets are taken into account.

Our guidelines can be viewed in the form of decision trees, available as supplementary material, and together with an interactive web interface on our website under the following link: https://cbdm. uni-mainz.de/ngs-guidelines. They can be used as a reference to classify the quality of a file. In the interactive view, a user can choose to filter for particular experimental conditions such as ChIP targets and evaluate the relevance of the quality features by comparing all files with the same targets. In addition, it is possible to observe which antibodies were used to target a protein, which in turn can be helpful to choose an antibody for a new experiment. Each of the 47 studied features, except the basic statistics from FastQC (RAW_Basic_statistics), can hold valuable quality information depending on the NGS experimental settings (e.g., assay,

biological sample or ChIP protein target). This is supported by the presence of a great variety of features in the different top 10 rankings in Fig 6 (21 out of 47 features). Some features do perform better overall although, especially the genome mapping (MAP) features, which are high-performing in statistical tests and were often selected to split nodes in the decision trees (Fig 6 and Table 1). Interestingly, there were multiple biological samples that were difficult to discriminate by quality using statistical tests. For example, in human paired-end transcription-factor ChIP-seq of HEK293 or HepG2 cells, none of the 47 features were significant (FDR > 0.05). Also, for human transcription-factor ChIP-seq, files related to HepG2 cells had no significant features for the PE files but four features were significant for the SE files. All quality features used in this study can be derived on new files using publicly available bioinformatics software (3, 5, 6, 19, 23).

The main limitations of our study are related to the relatively low number of available manually annotated low-quality files (n = 1,049), the imperfect matching of high-quality files in various subsets (only matched by organism, run type and assay), and the use of only one source of data, which is ENCODE. Because a perfectly balanced dataset may not substantially increase the number of studied data subsets or reduce observed biases, future work should focus on comprehensiveness of the guidelines in terms of metadata (e.g., species, assay, biological samples and ChIP protein and antibody targets). Because other resources such as Cistrome do not clearly classify their files for quality (e.g., offering a binary classification), they were not used to derive guidelines in this study. However, provided manual or accurate automatic processing, multiple online

resources may prove valuable in the future to extend the quality guidelines (15, 24, 25, 26).

Our study is also limited to functional genomics assays and to our selection of quality features, which were still able to discriminate accurately files by quality in various data subsets using decision trees or, in a previous work, more advanced machine-learning algorithms (3). Thus, additional assay-specific features may not contribute significantly to classification performance for those assays but may be instrumental for other NGS applications such as molecular diagnosis and variant detection (27, 28, 29). Compared to more advanced algorithms such as Random Forest, decision trees have the advantage of comprehensibility that is required here for offering practical guidelines, but the disadvantage to be unstable by high sensitivity to small changes in the training data or changes in the parameters (30). Therefore, our guidelines only offer one solution among others for the classification of a given set of NGS files. Selecting the most appropriate or representative tree may be achieved by a multi-criteria decision analysis–based process (31). Importantly, we enabled the estimation of the reliability of our results by providing statistics related to sample size, compositional bias, and also *P*-values for statistical tests.

Several considerations should be taken into account when using our guidelines. The low number of files of some of our subsets limits the reliability of related decision trees and statistical tests, although we tried to account for this by restricting our analyses to subsets with a minimum number of files. Large biases in the composition of the data subsets towards a single or few cell types or ChIP targets may call into question the guidelines, affecting the results of some statistical tests and decision trees. Biases can also prevent statistical tests from finding true differences between low- and high-quality files. Nevertheless, some decision trees may split the data explaining biases into dedicated branches. It should be mentioned that even if many of the trees could correctly divide the subsets by quality with only one feature, a user should always consider to observe the detailed distribution of values of the feature using the interactive dashboard and also consider other features. The most reliable results will be generated from data subsets where low- or high-quality files represent homogeneously a large variety of experimental conditions.

In the future, it will be interesting to build a database or augment an existing one with the possibility to label the data by quality. This could be done with manual and/or machine-learning algorithms and would help to overcome the current biases in the labelled sets of NGS experiments leading to newly derived guidelines. However, this will require careful manual work to ensure the highest standard of the derived guidelines. We cannot stress enough that an important contribution to this effort should be the deposition in the databases of low-quality data annotated as such by the laboratories that produce them.

As conclusion, we provided guidelines for the community to assess the quality of NGS data files and to better understand differences and relevance of quality features in specific experimental conditions. These guidelines can be used to evaluate already available but unlabeled data files as well as newly created ones. They can also be used for the planning of a new NGS experiment because they allow the identification of antibodies that have shown bad performance in the past. To allow further improvements of such guidelines, we recommend researchers to release publicly low-quality data and negative results. Publishers, databases, and funding agencies should think of mechanisms to encourage and reward such an activity, which is counter-intuitively beneficial for scientific development.

# Materials and Methods

### ENCODE dataset and quality features

Our main dataset was downloaded from ENCODE and contains 1,049 low-quality files found as of 21 November 2019, and 1,049 high-quality files randomly selected from the database to match low-quality files by assay category (RNA-seq, ChIP-seq, and DNase-seq), organism (human and mouse) and run type (SE- and PE) to ensure an overall balanced dataset (Fig 1A). Lower levels of details are not balanced (Fig 1B–I). ENCODE manually labels these files as either revoked or released, respectively. We derived four feature sets from different QC- or analysis tools (Fig 2 and Table S1).

Because FastQC is the most commonly used QC tool for NGS, we used its summary statistics in which 11 quality features are flagged as either pass, warning or fail. These features with their respective flags were used to define our first feature set RAW (raw-read statistics).

While these features are computed from the raw data, others require mapping the reads to a reference genome first. A tool commonly used for mapping is Bowtie2, which was chosen for its popularity and applicability to different assays. To derive biologically meaningful insights other mapping tools could work better. For example, splice-aware mappers such as HISAT2 or STAR (8, 23) would be a better choice for RNA-seq. Bowtie2 aligns reads to long reference genomes using an index method based on the Burrows–Wheeler Transform (6). The mapping statistics are used as quality features and describe the percentage of unmapped reads, as well as uniquely or multiply mapped reads and the overall mapping rate. These features are denoted as MAP (mapping-statistics features). In case of SE or PE files, those features are referred to as MAP_SE or MAP_PE, respectively. To allow specific comparison of single- and PE files, we created a set of MAP features called MAP_MI (mix between SE and PE). For single- or PE files, MAP_MI features are equal to either MAP_SE features or MAP_PE of concordant reads, respectively.

ChIPseeker implements a function to associate mapped reads with genomic features and is used to extract the third feature set, denoted as LOC (genomic localization features) (32). It is composed of nine features describing the distribution of reads mapped within promoter regions, first and following introns, 5′UTRs, first and following exons, 3′UTRs, distal intergenic regions and regions downstream proximal to the 3′UTR. ChipPeakAnno is used for the fourth feature set TSS, describing the distributions of reads near TSSs in the genome (33). Reads were counted in 10 bins defined within the 5 kb region up- and downstream of all TSS regions, resulting in 10 features identified by bin central relative coordinates (e.g., TSS_-4500 denoted the genomic region from −5 to −4 kb relative to the TSS regions).

To reduce RAM memory requirements during calculations, LOC and TSS features were computed from 1 million randomly sampled

mapped reads in each FastQ file. For paired-end files, the RAW features were derived independently for each of the two mate files, whereas MAP, LOC, and TSS features were derived for the pair of mate files itself. To reduce redundancy in the dataset, we filtered out the RAW features for the second mate of each pair. The largest files that were created in this step of data preprocessing are the FastQ and BAM files that have a summed size of 5.6 TB and 2 TB, respectively.

## The data subsets

The ENCODE dataset was divided into subsets of files organized in five groups (1, 2, A, B, and C) according to their experimental parameters. Groups 1 and 2 represent high-level subsets and are described in supplementary files, whereas lower level subsets (A, B, and C) are analyzed in the main text. Group 1 divides the data by organism (mouse or human), assay (e.g., ChIP-seq), and run type (single- or paired-end), resulting in six subsets, whereas group 2 uses assay title (e.g., histone ChIP-seq) instead of assay and results in 14 subsets. In the other groups, the total number of subsets in each subset group is 436 for subset group A, 354 for B, and 461 for C. There are 38 subsets each with at least 10 files in group A, apportioned by organism, run type, assay title, and biological sample. For example, one subset in group A contains 34 files all related to human paired-end DNase-seq in the thyroid gland. Group B consists of 41 subsets each with at least 10 files, apportioned by organism, run type, assay title and ChIP protein target. For example, a subset in group B contains 24 files all related to mouse single-end histone ChIP-seq of H3K4me3. The third group C adds the antibody to the criteria of group B and has 33 subsets of at least 10 files. For example, a subset in group C contains 16 files all related to mouse single-end transcription factor ChIP-seq of CTCF targeted with the antibody ENCAB210NHK (ENCODE antibody identifier). The full list of subsets is available in supplementary material: Supplementary_trees.pdf (Supplemental Data 1).

## Cistrome

We obtained the data for 47,201 files in the Cistrome database as of November 2020. Files were considered to be outliers if they had a FastQC sequence median quality score higher than 100 or a uniquely mapped reads ratio >1. A total of 235 outliers were identified and discarded, resulting in a dataset of 46,966 files. The quality features for all these files are provided by the Cistrome database as a csv file.

The following Cistrome's quality features denote high-quality files if the associated metrics are above the thresholds given as defined below:

- FASTQC: FastQCs raw sequence median quality score is >25%.
- UniquelyMappedRatio: The uniquely mapped read ratio computed with BWA is >60% (7).
- PBC: The PCR bottleneck coefficient, which is the number of locations with exactly one uniquely mapped read divided by the number of unique locations, is >80%.
- FRiP: The FRiP score, which is the fraction of uniquely mapped reads from autosomal chromosomes that are overlapping with MACS2 peaks, is >1% (19).

- PeaksUnionDHSRatio: The union DHS (DNase I hypersensitive sites) overlap, which is the proportion of the 5,000 most significant peaks that overlap with a union of DNase-seq peaks (derived by merging all DNAse-Seq data from ENCODE) is >70% (34).
- PeaksFoldChangeAbove10: Number of confident peaks (fold change >10) called by MACS2 is >500 (threshold derived from a plot on Cistrome's website) (34).

## Statistical analysis

Because we have both ordinal and quantitative features, and some of the quantitative features are not normally distributed (according to a Shapiro–Wilk test), we performed a Wilcoxon–Mann–Whitney test to assess the significance of the difference in the values of a feature between high- and low-quality files within each subset of files. To account for multiple testing, we calculated the FDR with the Benjamini–Hochberg method (35) within each group of subsets.

We also assessed the features' classification power by the area under Receiver Operating Characteristic curve (auROC). The feature values are used as predictors for the quality class and set against the true as well as the inverted quality vector. The greater of the two resulting auROCs is used as the classification power of the respective feature. Decision trees were computed using scikit-learn's CART algorithm.

## Plotting

All plots were produced in Python, with the help of the pandas, matplotlib, graphviz, and Seaborn packages (36, 37, 38, 39). Decision trees are part of the scikit-learn package (40).

# Data Availability

Public data are used in this study and available at the ENCODE project's website: https://www.encodeproject.org.

# Supplementary Information

# Acknowledgements

We thank Dr Steffen Albrecht for data collection and meaningful comments.

## Author Contributions

M Sprang: formal analysis, visualization, methodology, and writing—original draft, review, and editing.
M Krüger: formal analysis, visualization, and methodology.
MA Andrade-Navarro: supervision, funding acquisition, methodology, project administration, and writing—original draft, review, and editing.

J-F Fontaine: conceptualization, data curation, formal analysis, supervision, investigation, visualization, methodology, and writing—original draft, review, and editing.

## Conflict of Interest Statement

The authors declare that they have no conflict of interest.

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
