## [Reviewer comments · Life Science Alliance]

Life Science Alliance

Statistical guidelines for quality control of next generation sequencing techniques

Maximilian Sprang, Matteo Krüger, Miguel Andrade-Navarro, and Jean Fontaine

DOI: <https://doi.org/10.26508/lsa.202101113>

Corresponding author(s): Jean Fontaine, Johannes Gutenberg University of Mainz

Review Timeline:

Submission Date:	2021-05-05
Editorial Decision:	2021-07-22
Revision Received:	2021-08-11
Editorial Decision:	2021-08-16
Revision Received:	2021-08-17
Accepted:	2021-08-18

Scientific Editor: Novella Guidi

Transaction Report:

July 22, 2021

Re: Life Science Alliance manuscript #LSA-2021-01113-T

Dr. Jean-Fred Fontaine
Faculty of Biology, Johannes Gutenberg University Mainz, Hans-Dieter-Hüsch-Weg 15, 55128,
Mainz, Germany

Dear Dr. Fontaine,

Thank you for submitting your manuscript entitled "Statistical guidelines for quality control of next generation sequencing techniques" to Life Science Alliance. The manuscript was assessed by an expert reviewer, whose comment is appended to this letter. We also consulted with one of our academic editors member since at this time we have received only one reviewer report. As you will note from the comment below, the reviewer raises several concerns about the usage of the ENCODE datasets that are far from being reference-level (and suggests instead to use PCR-free libraries), about the analysis pipelines to map the reads which fails to provide robust and accurate mapping results (and suggests instead to use FANSe series algorithm) and also about the fact that the authors didn't evaluate the biological findings and didn't include the experimental part of NGS in the guideline. We, thus, encourage you to submit a revised version of the manuscript back to LSA that responds to all of the reviewers' points.

Thank you for this interesting contribution to Life Science Alliance. We are looking forward to receiving your revised manuscript.

Sincerely,

Novella Guidi, PhD

- A letter addressing the reviewers' comments point by point.
- An editable version of the final text (.DOC or .DOCX) is needed for copyediting (no PDFs).
- High-resolution figure, supplementary figure and video files uploaded as individual files: See our detailed guidelines for preparing your production-ready images, <https://www.life-science-alliance.org/authors>
- Summary blurb (enter in submission system): A short text summarizing in a single sentence the study (max. 200 characters including spaces). This text is used in conjunction with the titles of papers, hence should be informative and complementary to the title and running title. It should describe the context and significance of the findings for a general readership; it should be written in the present tense and refer to the work in the third person. Author names should not be mentioned.

B. MANUSCRIPT ORGANIZATION AND FORMATTING:

Reviewer #2 (Comments to the Authors (Required)):

The authors claimed a "guideline" for NGS quality control. "Guideline" is a big word which necessitate rigorous evidence. However, the authors evidence is far from "guideline".

1. The datasets need to be reference-level high quality. However, the ENCODE datasets are far from reference-level. There are numerous other high quality datasets available, in particular, the PCR-free libraries, which eliminates the PCR amplification bias. At least, the authors could use the reference datasets provided by the sequencer manufacturers.
2. The analysis pipelines are not the most accurate one. For example, the authors used Bowtie2 to map the reads. However, many evidence, especially the experimental evidence showed that

Bowtie2 failed to provide robust and accurate mapping results, which would mislead the subsequent analysis. FANSe series algorithm is a much more robust and accurate one to use. Please refer to : Wu et al., Journal of proteome research (2014), 13 (6), 2724-2734; Xiao et al., PLoS One (2014), 9(4):e94250; Mai et al., Scientific Reports (2017) 7:1053.

3. The purpose of NGS is to make biological relevant findings, e.g. mutations in DNA-seq, and gene expression level in RNA-seq. The goal of NGS is not to uniquely map more reads. The authors did not evaluate the biological findings. I strongly suggest the authors use the standard benchmarking dataset (Chen et al., Scientific Reports 10:3501).

4. The NGS is not just bioinformatics, bur rather wet lab procedures. A proper guideline must include the experimental part to ensure the production of high quality, low bias raw data. The statistical evaluation of experimental factors is essential.

We thank Reviewer #2 for their time and valuable points.

“1. The datasets need to be reference-level high quality. However, the ENCODE datasets are far from reference-level. There are numerous other high quality datasets available, in particular, the PCR-free libraries, which eliminates the PCR amplification bias. At least, the authors could use the reference datasets provided by the sequencer manufacturers.”

We agree with the reviewer that PCR-free datasets would eliminate a potential bias and that reference-level datasets are of great interest, but the goal of this work is to characterize the quality of routinely generated functional genomics NGS data in academic institutions to derive statistical quality guidelines. Such data is actually PCR-amplified and well represented by the ENCODE datasets. We have modified the manuscript to clarify this point (see changes in abstract and introduction). We want to mention that the main limitation in our study was the access to low-quality data (or data that is labeled as such). Furthermore, ENCODE has been used as reference in benchmarks before (yet differently from “reference-level” as mentioned by the reviewer): For example, both Kumar et al. *Nature Biotechnology* volume 31, pages 615–622 (2013) and Ampuja et al. *BMC Genomics*. 2017; 18: 68. use unions of ENCODEs peaks as benchmark. Cistrome, another database holding functional genomics data with quality information, uses a union of DNase-seq peaks from ENCODE as reference for one of their quality flags.

As the reviewer mentioned, Illumina provides some reference demo datasets in its BaseSpace online portal. Among them, only a few are related to functional genomics, and they reuse often the same universal reference samples but on different platforms. Such datasets may be interesting to compare sequencing platforms but they would not allow the generation of condition-specific guidelines for functional genomics NGS data files. Moreover, the following analysis demonstrates that samples data from the Illumina’s dataset also have different quality.

We compared 4 Illumina’s datasets by % intergenic reads with our respective Guideline LOC_Distal_intergenic (in human polyA+ RNA-seq high-quality files showed a mean around 7.9% and low-quality files at 12.37%). Other features available in BaseSpace were not directly comparable to our proposed features. For example, other LOC features were derived with another level of aggregation: UTR and Intron percentages are available in our features, but are parted into 3’ UTR and 5’ UTR as well as First Intron and Other Introns (See barplot below for some examples).

[Figure removed by editorial staff per authors' request].

The percentage of intergenic reads for samples of the selected datasets is reported in the following table:

Dataset	Assay Type	% intergenic reads
Project HiSeq 2000: TruSeq RNA Access (MAQC & Lung T/N)	TruSeq Total RNA	1.3% - 3.3% (8 samples)
Project NovaSeq 6000 SP	TruSeq Stranded mRNA	1.9% - 5.0% (10 samples)
Project NextSeq 500 v2: RNA-Seq (8plex)	TruSeq Stranded mRNA	4.0% - 4.8% (4 samples)
Project NextSeq 500 v2: RNA-Seq (8plex)	TruSeq Total RNA	13.9% - 16.7% (4 samples)

The 3 first rows seem of good quality. The last row seems of lower quality. Project related to the 2 last rows has data for the same brain and UHRR samples but generated using either stranded or total rna kits (see notes below). Moreover, the probable difference in quality as suggested by our guidelines is also reflected by the different number of detected genes at coverage 100X as follows.

Project NextSeq 500 v2: RNA-Seq (8plex):

Samples	Assay Type	N genes at 100X
2 Brain samples	TruSeq Stranded mRNA	5433 and 5300
2 Brain samples	Total RNA samples	4880 and 4678
2 UHRR samples	TruSeq Stranded mRNA	6254 and 6176
2 UHRR samples	Total RNA samples	5917 and 5152

=====
NOTES

=====

Project HiSeq 2000: TruSeq RNA Access (MAQC & Lung T/N)

RNA sequencing of total RNA isolated from FFPE samples or fresh-frozen samples prepared using the TruSeq RNA Access Kit. Differential expression analysis of tumor and normal tissue was performed for two human FFPE RNA samples: matched tumor and normal lung tissue derived from the same patient. Differential expression analysis was performed for two fresh-frozen RNA samples: MAQC Brain and Universal Human Reference RNA (UHR) samples. All samples were processed using two technical replicates.

Project NovaSeq 6000 SP: TruSeq Stranded mRNA (replicates of liver and UHRR)

Eight replicates each of UHRR and liver samples were prepared using the TruSeq Stranded mRNA kit and sequenced on a NovaSeq 6000 instrument with a SP flow cell at 2x51bp. Analysis was performed on BaseSpace using the RNA-Seq Alignment and DESeq2 apps.

Project NextSeq 500 v2: RNA-Seq (8plex)

Data generated from MAQC Human Brain Reference RNA (HBRR) and Universal Human Reference RNA (UHRR) sequenced on NextSeq 500 using V2 reagents. Samples prepared using TruSeq Stranded mRNA and Total RNA reagent kits.

“2. The analysis pipelines are not the most accurate one. For example, the authors used Bowtie2 to map the reads. However, many evidence, especially the experimental evidence showed that Bowtie2 failed to provide robust and accurate mapping results, which would mislead the subsequent analysis. FANSe series algorithm is a much more robust and accurate one to use. Please refer to: Wu et al., Journal of proteome research (2014), 13 (6), 2724-2734; Xiao et al., PLoS One (2014), 9(4):e94250; Mai et al., Scientific Reports (2017) 7:1053.”

Our pipeline is a generic pipeline that can be applied to various types of NGS files (RNA-seq, DNase-seq and ChIP-seq) for the main purpose of comparing the results to our guidelines for a decision-making process about the quality of a NGS file. We have updated the manuscript to clarify that other tools are likely to be more relevant to derive biological insights depending on the experimental settings (see Page 3). For example, splice aware alignment tools such as the HISAT2 or STAR will be more relevant for RNA-seq assays.

The tools we chose for feature generation were chosen for their popularity (community validation) and/or free accessibility. Otherwise, our data-driven guidelines could not to be used by a broad community. Bowtie2 is free software, which was cited 18939 times as of 26.07.2021, so it complies with those critical points. FANSe is commercial software, which was cited less (FANSe: 48, FANSe2: 41, FANSe3: 2 citations) and therefore does not comply with those critical points.

Furthermore, an alleged problem with Bowtie2's mapping would impact all analyzed samples, which in turn would not impact the quality analysis we provide here, since it is built upon significant differences between high- and low-quality samples.

“3. The purpose of NGS is to make biological relevant findings, e.g. mutations in DNA-seq, and gene expression level in RNA-seq. The goal of NGS is not to uniquely map more reads. The authors did not evaluate the biological findings. I strongly suggest the authors use the standard benchmarking dataset (Chen et al., Scientific Reports 10:3501).”

The aim of this work is to simplify a decision-making procedure about the quality of a fastq file. By allowing scientists to better assess the quality of their NGS files, we assume that our data-driven guidelines will have a positive impact on their findings.

The work referenced by the reviewer; “Systematic comparison of somatic variant calling performance among different sequencing depth and mutation frequency” Chen et al 2020; was about somatic variant calling, which is not in the scope of our work, since we are concerned here with functional genomics assays.

“4. The NGS is not just bioinformatics, but rather wet lab procedures. A proper guideline must include the experimental part to ensure the production of high quality, low bias raw data. The statistical evaluation of experimental factors is essential.”

In the introduction of our manuscript, we discuss the ENCODEs guidelines as a source for wet-lab guidelines. We also position our work as complementary since we aim to present statistical guidelines meant as a help for researchers to easily assess the quality of a given data file, not to produce such data.

August 16, 2021

RE: Life Science Alliance Manuscript #LSA-2021-01113-TR

Dr. Jean Fred Fontaine
Johannes Gutenberg University of Mainz
Hans-Dieter-Hüsch-Weg 15
Mainz, RHEINLAND-PFALZ 55128
Germany

Dear Dr. Fontaine,

Thank you for submitting your revised manuscript entitled "Statistical guidelines for quality control of next generation sequencing techniques". We would be happy to publish your paper in Life Science Alliance pending final revisions necessary to meet our formatting guidelines.

- please add ORCID ID for the corresponding author-you should have received instructions on how to do so
- please upload your Tables in editable .doc or excel format
- please upload your main and supplementary figures as single files
- please consult our manuscript preparation guidelines <https://www.life-science-alliance.org/manuscript-prep> and make sure your manuscript sections are in the correct order;
- please add your main, supplementary figure, and table legends to the main manuscript text after the references section
- please add an Author Contributions section to your main manuscript text
- please add callouts for Figure 1B-I to your main manuscript text

LSA now encourages authors to provide a 30-60 second video where the study is briefly explained. We will use these videos on social media to promote the published paper and the presenting author. Corresponding or first-authors are welcome to submit the video. Please submit only one video per manuscript. The video can be emailed to contact@life-science-alliance.org

A. FINAL FILES:

B. MANUSCRIPT ORGANIZATION AND FORMATTING:

Sincerely,

August 18, 2021

RE: Life Science Alliance Manuscript #LSA-2021-01113-TRR

Dr. Jean Fred Fontaine
Johannes Gutenberg University of Mainz
Hans-Dieter-Hüsch-Weg 15
Mainz, RHEINLAND-PFALZ 55128
Germany

Dear Dr. Fontaine,

Thank you for submitting your Research Article entitled "Statistical guidelines for quality control of next generation sequencing techniques". It is a pleasure to let you know that your manuscript is now accepted for publication in Life Science Alliance. Congratulations on this interesting work.

DISTRIBUTION OF MATERIALS:

Again, congratulations on a very nice paper. I hope you found the review process to be constructive and are pleased with how the manuscript was handled editorially. We look forward to future exciting submissions from your lab.

Sincerely,
